# RLGF: Reinforcement Learning with Geometric Feedback for Autonomous Driving Video Generation

**Tianyi Yan**[*]
SKL-IOTSC, CIS
University of Macau
tianyi.yan123@gmail.com

**Wencheng Han**[*]
SKL-IOTSC, CIS
University of Macau

**Xia Zhou**
Li Auto Inc.

**Xueyang Zhang**
Li Auto Inc.

**Kun Zhan**
Li Auto Inc.

**Cheng-zhong Xu**
SKL-IOTSC, CIS
University of Macau

**Jianbing Shen** [†]
SKL-IOTSC, CIS
University of Macau

## Abstract

Synthetic data is crucial for advancing autonomous driving (AD) systems, yet current state-of-the-art video generation models, despite their visual realism, suffer from subtle geometric distortions that limit their utility for downstream perception tasks. We identify and quantify this critical issue, demonstrating a significant performance gap in 3D object detection when using synthetic versus real data. To address this, we introduce Reinforcement Learning with Geometric Feedback (RLGF), RLGF uniquely refines video diffusion models by incorporating rewards from specialized latent-space AD perception models. Its core components include an efficient Latent-Space Windowing Optimization technique for targeted feedback during diffusion, and a Hierarchical Geometric Reward (HGR) system providing multi-level rewards for point-line-plane alignment, and scene occupancy coherence. To quantify these distortions, we propose GeoScores. Applied to models like DiVE on nuScenes, RLGF substantially reduces geometric errors (e.g., VP error by 21%, Depth error by 57%) and dramatically improves 3D object detection mAP by 12.7%, narrowing the gap to real-data performance. RLGF offers a plug-and-play solution for generating geometrically sound and reliable synthetic videos for AD development.

## 1 Introduction

The rapid progress of autonomous driving (AD) systems [15, 17, 6] has created a growing need for high-quality synthetic data. Recent diffusion-based video generation methods [57, 61, 30, 10, 9, 18, 19] have achieved state-of-the-art visual realism, measured by metrics like FVD [49]. However, we identify a critical yet underexplored limitation: the generated videos often contain subtle yet impactful incorrect geometric relationships. This flaw not only misleads downstream perception and planning tasks but also undermines the reliability of models trained or evaluated using such data, significantly constraining their applicability in essential use cases such as simulation-based training [30] and system validation [61, 67].

To investigate this geometric limitation, we conduct a series of targeted experiments. Evaluating 3D object detection on synthetic videos using BEVFusion [26] reveals a substantial performance drop compared to real data (mAP: 25.7 vs. 35.5). In contrast, 2D object detection using YOLOv5 [20]

---

[*]Equal contribution
[†]Corresponding author

39th Conference on Neural Information Processing Systems (NeurIPS 2025).

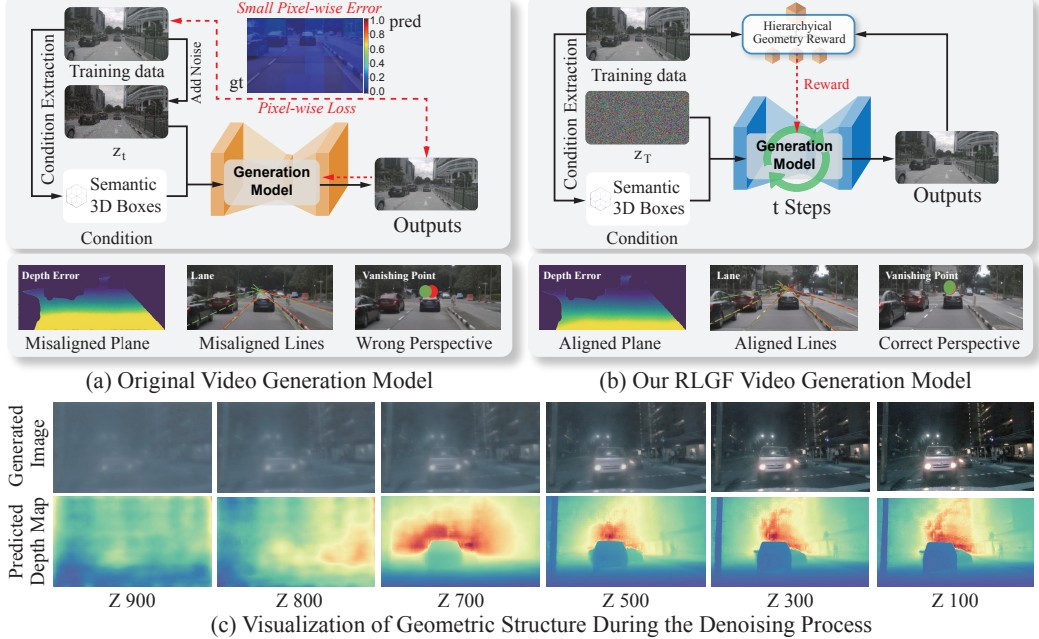

(a) Original Video Generation Model  (b) Our RLGF Video Generation Model

(c) Visualization of Geometric Structure During the Denoising Process

Figure 1: (a) **Original Video Generation Models**, optimized via pixel-level supervision (e.g., noise prediction error), often produce visually plausible videos that nonetheless suffer from severe geometric flaws (misaligned planes/lines, wrong perspective). This can degrade downstream tasks like 3D object detection (e.g., mAP drop from 35.5 to 25.7). (b) **Our RLGF** integrates a Hierarchical Geometry Reward directly into the multi-step denoising process. This reward, derived from perception models, guides the generation model to produce outputs with aligned planes, correct lane structures, and accurate perspective. (c) **Visualized depth maps** from noisy latents at various denoising stages (from noisy to less noisy) show coarse geometry emerging early and details later. This motivates our Latent-Space Windowing Optimization for targeted intermediate rewards.

on the same data yields results comparable to real-world samples (mAP: 43.8 vs. 44.7). These findings suggest that while current diffusion models [57, 56, 18, 9] preserve 2D appearance, indicating minimal image-level domain gap, yet they fail to capture accurate 3D scene structure. We attribute this primarily to underlying geometric inconsistencies. To further verify this hypothesis and systematically quantify these distortions, we introduce GeoScores. This metric suite evaluates geometric fidelity by applying pre-trained perception models [65, 42] to both synthetic videos and their corresponding real-world counterparts, using the outputs from real videos as reference ground truth. Significant discrepancies between the two highlight geometric errors. GeoScores (details are in appendix A.1) reveals three major issues: (1) Vanishing point shifts, indicating incorrect global perspective; (2) Lane topology inconsistencies, reflecting misaligned road markings and implausible lane structures; and (3) Depth errors, particularly on road surfaces, signifying incorrect placement. (e.g., for a typical baseline, an average VP shift of 0.086 normalized units, a Lane F1-Score of only 0.792, and an average depth error of 1.822). The significant deviations in these scores confirm that current "high-quality" synthetic data often suffers from pervasive geometric inaccuracies (fig. 1(a)).

Addressing these challenges, we present Reinforcement Learning with Geometric Feedback (RLGF), a novel framework that injects perception-model-driven geometric spatial constraints directly into the video generation process. Unlike conventional approaches [57, 56, 61, 18] that primarily rely on pixel-wise alignment, which often fail to explicitly enforce adherence to complex, underlying geometric principles, RLGF leverages dedicated, pre-trained AD perception models as reward providers to ensure geometric fidelity.

RLGF introduces two core technical innovations: Latent-Space Windowing Optimization and Hierarchical Geometric Alignment. Frist, we present Latent-Space Windowed Optimization. We observe that geometric structures in diffusion models emerge progressively across denoising steps: early steps

(e.g., before step 10 in flow matching [27, 28]) establish coarse global geometry, while later steps refine local details (fig. 1(c)). Training across the entire sampling process, as done in some prior RL-diffusion work [1] struggles to provide targeted guidance for these distinct phases. Therefore, we propose an efficient latent-space training strategy where rewards are applied directly to noisy latent features within a randomly sampled sliding window of intermediate diffusion steps. This approach significantly reduces computational (GPU memory) burden and, more importantly, allows for effective and targeted corrective feedback during both early (global structure formation) and late (detail refinement) stages of geometric synthesis.

Secondly, RLGF features Hierarchical Geometric Reward (HGR), a multi-level feedback system designed to imbue generated videos with robust geometric fidelity and scene coherence. HGA integrates signals from two specialized latent-space perception networks: a Latent Geometry Perception Model assessing vanishing point, lane, and depth cues, and a Latent Occupancy Prediction Model inferring 3D scene occupancy. Both operate efficiently on noisy latents, aided by a lightweight micro-decode module to circumvent costly full decoding. Leveraging these perception models, HGA then constructs its multi-level reward system. For point-line-plane geometric feedback, we use the outputs of the perception model to: (1) enforce vanishing point consistency for accurate global perspective, (2) ensure lane topology validity for realistic road structure, and (3) promote depth coherence for correct surface and object geometry. For scene-level occupancy feedback, outputs from the occupancy model are used to: (4) align intermediate semantic features using KL divergence for plausible scene evolution, and (5) maximize 3D occupancy IoU for accurate volumetric object layout and dynamics. This structured approach, combining efficient latent-space perception with a multi-faceted reward design, ensures that comprehensive geometric and occupancy information is extracted and effectively fed back to guide the video diffusion model towards producing physically principled and geometrically sound autonomous driving scenarios.

Our contributions are fourfold:

- We are the first to systematically quantify the geometric distortion problem in autonomous driving video generation and propose the GeoScores metric for its evaluation.

- We introduce RLGF, a novel paradigm that uses reinforcement learning with perception-based rewards applied efficiently within a sliding window in latent space, enabling plug-and-play geometric correction.

- We design HGR which addresses geometric distortions by incorporating point-line-plane and scene-level occupancy multi-level geometric feedback derived from latent representations.

- Extensive experiments on nuScenes demonstrate RLGF's plug-and-play effectiveness across two baselines [18, 9], boosting 3D detection mAP by 12.7% absolute while reduce geometry gap (via GeoScores) relative to real data. This work establishes a new paradigm for geometrically faithful synthetic data generation in autonomous driving systems.

## 2 Related Work

### 2.1 Video Diffusion for Autonomous Driving

The development of robust autonomous driving (AD) systems [15, 17, 6] necessitates large volumes of diverse and realistic training data. Learned generative model [22, 62, 11] have emerged as a powerful approach to synthesize such data by capturing the complex distributions of real-world driving scenarios. Early efforts explored Generative Adversarial Networks (GANs)[11, 38] and Variational Autoencoders (VAEs) [21] for generating driving-related imagery. More recently, diffusion models [12, 35, 44, 39, 59, 69, 68, 8], including latent diffusion models [44], have demonstrated state-of-the-art capabilities in generating high-fidelity images and videos [44, 4, 3, 77, 51, 54, 55, 75, 34]. Based on this technique, generative models in autonomous driving have greatly advanced. For example, BEVGen [46] and BEVControl [64] generate controllable street-view imagery from Bird's-Eye View (BEV) layouts or other structural conditions. Further advancements, such as DriveDreamer [53], Magicdrive [10],DriveWM [56], Panacea [57], DriveSphere [61] and others [76, 30, 31, 58, 18, 23, 32, 73, 72], focus on generating coherent multi-camera driving videos, often conditioned on textual prompts, historical trajectories, or 3D assets. These models excel at producing visually compelling outputs that achieve high scores on perceptual metrics like Fréchet Video Distance (FVD) [49].

Despite achieving high visual realism, leading AD video generation models [57, 9, 18] frequently introduce subtle geometric and scene-level distortions (e.g., flawed perspectives, depth errors, unrealistic motion). These inconsistencies, often overlooked by human assessment, severely undermine performance on 3D perception tasks like object detection [26] and motion forecasting. RLGF addresses this critical gap by proposing a novel framework to instill multi-scale geometric and scene-level consistency within the generation process.

## 2.2 Reinforcement Learning for Video Generation

Fine-tuning generative models with reward signals, rooted in Reinforcement Learning from Human Feedback (RLHF) successes in LLMs (e.g., using PPO [45] or DPO [41]), is increasingly applied to diffusion models. For image generation, methods [2, 50, 63, 60] align models with human preferences, and approaches [25] use perception model feedback for content control. This trend extends to video diffusion, where DPO-based techniques [66, 74, 71] often utilize human preference data. While improving general video quality, these holistic preference scores typically lack the precise, local geometric feedback crucial for autonomous driving applications. Other methods [1, 24, 33, 7] employ explicit reward models for video fine-tuning, offering more detailed signals. However, these rewards are generally not tailored to the specific multi-scale 3D geometric and physical plausibility demands of AD simulation, failing to adequately address distortions like incorrect perspective or depth error.

Our RLGF framework advances this by introducing explicit, quantifiable geometric and scene-level rewards from dedicated AD perception models. This provides targeted, locally-aware feedback to correct specific geometric inaccuracies, producing data suitable for rigorous AD tasks.

# 3 Method

## 3.1 Preliminary: Limitations of Conditional Video Diffusion

Current video diffusion models [3, 57, 18] generate frames by gradually denoising latent variables through a Markov chain. Formally, given a condition $\mathbf{c}$ (e.g., road sketches or bounding boxes), the model learns to minimize the pixel-level reconstruction error during training:

$$\mathcal{L}_{\text{pixel}} = \mathbb{E}_{x_0, \epsilon, t}[||\epsilon - \epsilon_\theta(x_t, t, c)||^2], \tag{1}$$

where $x_t$ is the noisy sample at timestep $t$, $\epsilon$ is the ground truth noise, and $\epsilon_\theta$ is the denoising network. While effective for visual fidelity, this standard formulation inherently ignores crucial geometric-semantic correlations due to two limitations: (1) Pixel-wise Independence Assumption: The MSE loss treats each pixel as independent, failing to model higher-order geometric relationships (e.g., perspective consistency in 3D space). (2) Conditional Oversimplification: Existing methods concatenate geometric conditions $c$ with noisy latents, However, this primarily enforces local pixel alignment corresponding to the condition rather than guaranteeing the global geometric integrity or plausibility of the underlying 3D scene structure. These limitations motivate the need for explicit geometric guidance during generation.

## 3.2 Reinforced Learning with Geometric Feedback (RLGF)

Our primary objective is to enhance the geometric and spatio-temporal consistency of videos generated by conditional diffusion models, addressing the critical gap left by conventional training objectives that primarily focus on pixel-level visual fidelity.

To achieve this, we introduce Reinforcement Learning with Geometric Feedback (RLGF). This framework refines a pre-trained video diffusion model, $\epsilon_\theta$, by guiding its generation process towards outputs that exhibit greater adherence to real-world geometric and physical principles. Unlike RLHF [37, 63], which typically captures broad subjective preferences, RLGF is designed to incorporate specific, model-interpretable geometric constraints. The core idea is to leverage dedicated, pre-trained perception models to evaluate the generated videos $x_0$ and provide a reward signal that steers $\epsilon_\theta$ away from implausible outcomes.

Formally, given a well-trained video diffusion model $\epsilon_\theta$, the dataset $D_v$, the well-designed reward function $R(\cdot)$, we aim to maximize the objective:

$$J(\theta) = \mathbb{E}_{c, v, x_0}[R(x_0, v)] \tag{2}$$

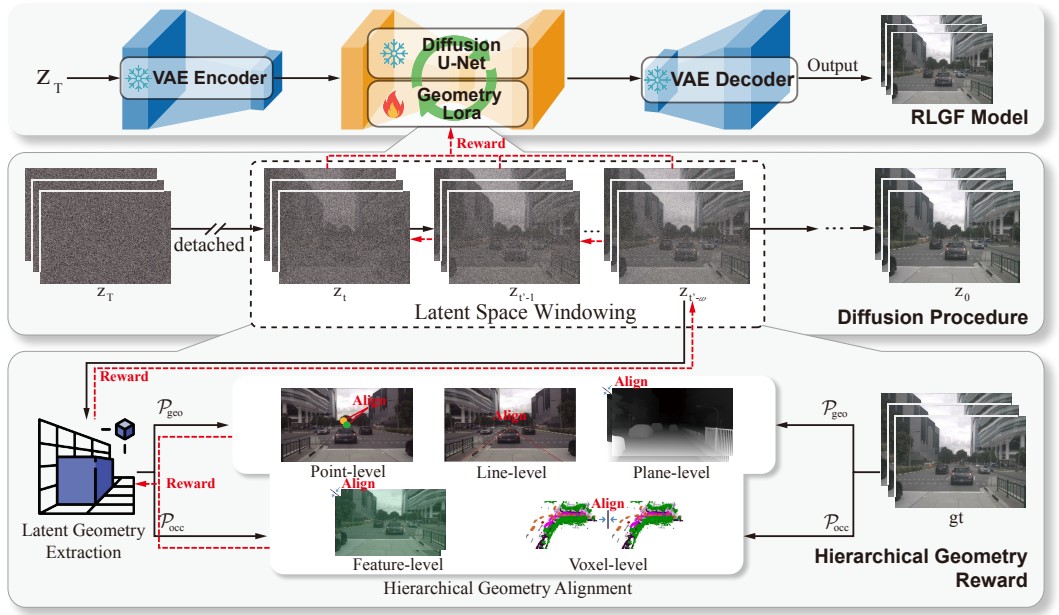

Figure 2: **Overview of the Reinforcement Learning with Geometric Feedback (RLGF) framework.** RLGF fine-tunes a frozen well-trained diffusion model via LoRA using rewards from a "Latent Space Windowing" scheme. Within this window, intermediate latents $z_{t'-w}$ are evaluated by frozen perception models $\mathcal{P}_{geo}$ (point-line-plane alignment) and $\mathcal{P}_{occ}$ (scene-level consistency) against a reference video. The resulting rewards ($R_{geo}, R_{occ}$) generate gradients (red arrows) to update LoRA, improving geometric and temporal consistency. Black arrows: feed forward; dashed red: gradients.

where $c$ and $v$ denote the condition and real video from the dataset $D_v$, $x_0$ the the generated sample by $\epsilon_\theta(\cdot|c)$ given $c$, and the total reward $R(x_0, v)$ encapsulates assessments from our Hierarchical Geometric Alignment system, as detailed in Section 3.4.

## 3.3 Latent-Space Windowing RL Optimization

Directly optimizing Equation 2 by unrolling the entire $T$-step diffusion sampling process to obtain $x_0$ and then backpropagating the reward gradient $\nabla_\theta R(x_0, v)$ is computationally prohibitive due to large memory requirements and long sampling chains.

To efficiently apply RLGF, we propose Latent-Space Windowed RL Optimization. This strategy addresses the computational challenge of full rollouts and leverages insights into the progressive nature of structure formation in diffusion models.

Visualizations (e.g., fig. 2(c) in Introduction) suggest that coarse global geometry is often established in earlier denoising steps (e.g. $t > T_{mid}$), while later steps refine details. Training across the entire sampling chain, as in some prior RL-diffusion work [1], may not provide sufficiently targeted guidance for these distinct phases.

Instead of rewarding only the final output $x_0$, we provide feedback based on noisy latent features $z_t$ at intermediate denoising steps $t$, compared against reference latents $z_v$ derived from real videos. Specifically, during the $T$-step sampling process $z_T \rightarrow \cdots \rightarrow z_0$, we apply our reward functions $R$ within a sliding window of $w$ steps (e.g. from random start step $t'$ down to $k = t' - w$). Our perception models are designed to take $z_k$ and $k$ as input.

This approach is motivated by two factors: (1) Efficiency: It significantly reduces the computational graph for backpropagation. (2) Effectiveness: It allows targeted corrective signals at different stages of generation, crucial for both global structure and local detail.

The practical objective then becomes maximizing the expected reward obtained by evaluating the noisy latent $z_k$ (at step $k$ within the window) against the VAE-encoded latent $z_v$ paired with real

video $v$:

$$J_{practical}(\theta_{LoRA}) = \mathbb{E}_{c,v,z_k}[R(z_k, z_v)] \tag{3}$$

where and $z_v$ is the encoded latent feature using VAE ecnoder [44]. The gradient with respect to the LoRA parameters $\theta_{LoRA}$ for a reward at step $k$ is then:

$$\nabla_{\theta_{LoRA}} R_k(z_k, z_v) = \frac{\partial R_k(z_k, z_v)}{\partial z_k} \cdot \frac{\partial z_k}{\partial \theta_{LoRA}} \tag{4}$$

Throughout the RLGF fine-tuning, the perception models that constitute the reward function are pre-trained and their weights remain frozen. ensuring a consistent evaluation standard as $\epsilon_\theta$ via $\theta_{LoRA}$ adapts to maximize geometric fidelity.

### 3.4 Hierarchical Geometric Reward (HGR)

The Hierarchical Geometric Reward ($R$) is designed to provide comprehensive, multi-level feedback on the geometric integrity and scene coherence of generated video latents. It evaluates consistency across point, line, plane, perceptual feature, and voxel-level representations, derived by applying specialized perception models to both the generated latent $z_k$ and the reference real-video latent $z_v$.

#### 3.4.1 Efficient Latent-Space Perception

To avoid computationally expensive full VAE decoding at each step $k$ for perception, our perception models operate directly on latent features. Given a noisy video latent $z_k \in \mathbb{R}^{L \times C \times H' \times W'}$ (where $L$ is frames, $C$ channels, $H', W'$ latent dimensions) at diffusion step $k$, we first employ a lightweight Micro-Decode Module, $\mathcal{F}_{micro}$. This module, constructed using shallow layers from the VAE decoder, processes $z_k$ and $k$ to produce enhanced per-frame features:

$$\mathbf{f}_k^f = \mathcal{F}_{micro}(z_k^f, k) \tag{5}$$

where $k$ is processed by the Fourier Embedding [77, 48]. $\mathbf{f}_k^f$ is suitable for downstream perception tasks, significantly reducing memory and computation. The same $\mathcal{F}_{micro}$ is applied to $z_v$ (with $k = 0$) to obtain reference features $\mathbf{f}_v^f$.

#### 3.4.2 Perception Model Architectures

We train two models that take these micro-decoded features $\{\mathbf{f}_k^f\}$ or $\{\mathbf{f}_v^f\}$ and timestep $k$ as input.

**Latent Geometry Perception Model** ($\mathcal{P}_{geo}$)**:** This multi-task model processes per-frame features $\mathbf{f}^f$ to assess static 2.5D geometry. It uses a DINOv2 [36] backbone followed by task-specific heads:

(1) For **vanishing point detection**, we reformulate the task as heatmap regression to better capture positional uncertainty, following [13]. The network head takes $\mathbf{g}$ as input and predicts a probability heatmap $H \in \mathbb{R}^{h \times w}$. The ground truth heatmap $H^{gt}$ is constructed as a 2D Gaussian centered at the annotated VP location, balancing localization precision and learning difficulty. We optimize using MSE:

$$\mathcal{L}_{vp} = \|H - H^{gt}\|^2 \tag{6}$$

(2) The **lane parsing** branch complements this by detecting road markings through a topology-aware segmentation head, following [42].

(3) As for the **depth estimation** task, we fine-tune the weight of [65] using a scale-invariant logarithmic (SiLog) loss:

$$\mathcal{L}_{depth} = \frac{1}{n}\sum_i d_i^2 - \frac{\lambda^2}{n}(\sum_i d_i)^2 \tag{7}$$

where $d_i = \log y_i - \log y_i^{gt}$, $y_i$ is the predicted results and $\lambda \in [0, 1]$

Finally, we optimize $P_{geo}$ using the total loss $\mathcal{L}_{geo} = \mathcal{L}_{vp} + \mathcal{L}_{lane} + \mathcal{L}_{depth}$.

**Latent Occupancy Prediction Model** ($\mathcal{P}_{occ}$)**:** This model processes per-frame features $\mathbf{f}^f$ using an image backbone followed by a prediction head to infer the intermediate backbone features $\mathbf{feat}_{occ}$ and the 3D occupancy grid $O^i \in \mathbb{R}^{X \times Y \times Z}$ representing the occupied space around the ego vehicle for that frame. $\mathcal{P}_{occ}$ is trained based on [16].

### 3.4.3 Multi-Granularity Reward Signals

The hierarchical geometric reward $R = R_{geo} + R_{occ}$ quantifies geometric fidelity and scene coherence through a multi-scale decomposition.

For $R_{geo}$, it addresses vanishing point consistency, lane topology validity, and depth coherence. Given the outputs $\{p_{vp}, L_{pred}, D_{pred}\}$ from $\mathcal{P}_{geo}(z_k)$ and reference conditions $\{v_{ref}, L_{ref}, D_{ref}\}$ from $\mathcal{P}_{geo}(z_v)$, we define:

$$R_{geo}(z_k, z_v) = \underbrace{\lambda_{vp} r_{vp}(p_{vp}, v_{ref})}_{\text{Point- level}} + \underbrace{\lambda_{lane} r_{lane}(p_{lane}, L_{ref})}_{\text{Line- level}} + \underbrace{\lambda_{depth} r_{depth}(p_{depth}, D_{ref})}_{\text{Plane-level}} \quad (8)$$

where $p_{vp}, p_{vp}, p_{vp}$ are the predictions from $\mathcal{P}_{geo}$ for vanashing point, lanes and depth, respectively. $\lambda_{vp}, \lambda_{vp}$ and $\lambda_{vp}$ weights balancing the contribution of each task. The individual reward functions $r(\cdot)$ are designed to be high for geometrically accurate samples and low (or negative) for inconsistent ones:

$$r_{vp}(p_{vp}, v_{ref}) = -\|p_{vp} - v_{ref}\|_2^2 \quad (9)$$

where $v_{ref}$ is the ground-truth vanishing point in conditions.

$$r_{lane} = \text{F1-Score}(L_{pred}, L_{ref}) \quad (10)$$

where F1-Score indicates the F1-Scores.

$$r_{depth} = -(\sqrt{(D_{pred} \odot M_{road} - D_{ref} \odot M_{road})^2} + \sqrt{(D_{pred} \odot M_{vehicle} - D_{ref} \odot M_{vehicle})^2})$$
$$(11)$$

where pixel masks $M_{road}$ and $M_{vehicle}$ are generated from real data and identify road and vehicle regions, respectively.

After that, we define $R_{occ}(z_k, z_v) = r_{align} + r_{iou}$. $r_{align}$ encourages similarity in high-level scene interpretation by aligning distributions of intermediate occupancy features. For each frame $f$:

$$r_{align_f} = -D_{KL}(p(\mathbf{feat}_{occ,f}^{real})\|p(\mathbf{feat}_{occ,f}^{gen})), \quad (12)$$

where $\mathbf{feat}_{occ,f}$ are backbone features from $\mathcal{P}_{occ}$. Distributions $p(\cdot)$ can be estimated (e.g., Gaussian fit over a batch).

$r_{occ}$ promotes accurate 3D structure and object layout. For each frame $f$:

$$r_{iou_f} = \text{IoU}(O_f^{gen}, O_f^{real}) = \frac{|O_f^{gen} \cap O_f^{real}|}{|O_f^{gen} \cup O_f^{real}|}, \quad (13)$$

where $O_f$ are the 3D occupancy grids from $\mathcal{P}_{occ}$.

## 4 Experiments

### 4.1 Experimental Setup

We conduct comprehensive experiments to validate the efficacy of our proposed Reinforcement Learning with Geometric Feedback (RLGF) framework. This section details the datasets, baseline models, evaluation metrics, and implementation specifics used in our evaluation.

**Datasets.** Our experiments are primarily conducted on nuScenes [5] using the official validation split. Its multi-camera setup and comprehensive annotations, including 3D object labels and HD maps, provide a challenging benchmark. For this dataset, conditions $c$ for the diffusion models are extracted from ground truth annotations, simulating realistic inputs for controllable generation.

**Baselines.** We demonstrate the plug-and-play nature of RLGF by integrating it with two representative video diffusion models, referred to as MagicDrive-V2 [9] and DiVE [18].

**Evaluation Metrics. GeoScores:** We use our proposed GeoScores suite: Vanishing Point Error: NormDist between the predicted VP and the pseudo-ground truth VP derived from real data. Lower is better. Lane Topology Score: F1-score for semantic segmentation of lane markings against ground

Table 1: **Performance of our Latent Geometry Perception Model ($\mathcal{P}_{geo}$) on the nuScenes validation split.** $\mathcal{P}_{geo}$ operates directly on micro-decoded latent features and is compared against representative pixel-space baselines.

| Task | Metric | Model / Method | Input Space | Performance |
|---|---|---|---|---|
| VP Detection | NormDist ↓ | URVP [29] | Pixel | 0.045 |
| | | VPD [14] | Pixel | 0.032 |
| | | $\mathcal{P}_{geo}$ | **Latent** | **0.024** |
| Lane Parsing | F1-Score ↑ | LaneATT [47] | Pixel | 0.822 |
| | | PriorLane [40] | Pixel | 0.879 |
| | | $\mathcal{P}_{geo}$ | **Latent** | 0.865 |
| Depth Estimation | RMSE ↓ | DepthAnything-v2 [65] | Pixel | **1.798** |
| | | $\mathcal{P}_{geo}$ | **Latent** | 2.596 |

Table 2: **Comparison with state-of-the-art video generation methods on the nuScenes validation set.** We evaluate visual quality (FVD), 3D Object Detection (3DOD) performance, and geometric fidelity (GeoScores).

| Methods | Quality | 3DOD | | GeoScore | | |
|---|---|---|---|---|---|---|
| | FVD | mAP | NDS | VP | Lane | Depth |
| Real Data | - | 35.53 | 41.20 | - | - | - |
| Panacea [57] | 139.0 | 11.58 | 22.31 | - | - | - |
| Drive-WM [57] | 122.7 | 20.66 | - | - | - | - |
| MagicDrive-v2 [18] | 101.2 | 18.95 | 21.10 | 0.092 | 0.787 | 1.732 |
| DiVE [18] | 68.4 | 25.75 | 33.61 | 0.086 | 0.792 | 1.822 |
| MagicDrive-v2+Ours | 99.8 | 23.21 | 27.80 | 0.079 | 0.854 | 0.983 |
| DiVE+Ours | **67.6** | **31.42** | **36.07** | **0.068** | **0.879** | **0.772** |

truth lane masks. Higher is better. Depth Error: We use RMSE between predicted depth for road surface regions and pseudo-ground truth depth. Lower is better.

**Downstream Task Performance: 3D Object Detection:** We employ a strong BEV-based detector, BEVFusion [26], trained solely on synthetic data generated by different methods or on real data. We report the standard nuScenes detection score (NDS) and mean Average Precision (mAP).

**Visual Realism:** We report Fréchet Video Distance (FVD) [49] to ensure RLGF does not degrade the visual quality achieved by the baseline diffusion models.

**Implementation Details** are included in the supplemental materials.

## 4.2 Performance of Pre-trained Perception Models

We first verify the capabilities of our perception models ($\mathcal{P}_{geo}$ and $\mathcal{P}_{occ}$ which form the basis of our reward functions. These models operate directly on latent features $z_t$ and timestep $t$. Our goal here is to demonstrate their competence in extracting meaningful geometric and scene information from the latent domain.

**Latent Geometry Perception Model ($\mathcal{P}_{geo}$)** is a multi-task model responsible for assessing fine-grained geometric properties from latent features. Table 1 presents its performance on the nuScenes validation split for vanishing point (VP) detection, lane parsing, and depth estimation, compared against established pixel-space methods. The results indicate that $\mathcal{P}_{geo}$ effectively captures these geometric cues from latent features. For instance, it achieves a normalized distance error of 0.024 for VP detection and an F1-score of 0.865 for lane parsing. Although its depth estimation RMSE (2.596) is higher than the specialized pixel space model DepthAnything-v2, it still provides a consistent measure of depth relationships.

**Latent Occupancy Model ($\mathcal{P}_{occ}$)** is tasked with understanding 3D scene layout from sequences of latent features. Its performance on the Occ3D-nuScenes is shown in Table 3. $\mathcal{P}_{occ}$ achieves an overall mIoU of 29.96 when operating from latent representations. This level of performance, compared to a pixel-based method like FlashOcc (32.08 mIoU), demonstrates a solid capability to infer volumetric scene structure from the latent domain.

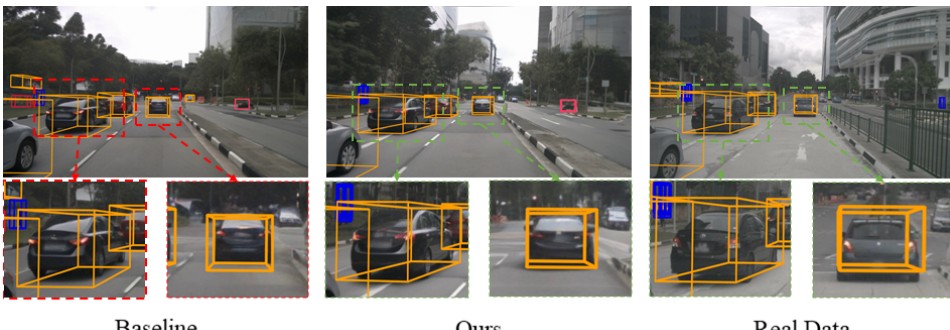

Baseline         Ours         Real Data

Figure 3: **Qualitative comparison of 3D object bounding box alignment.** RLGF-enhanced video exhibits much-improved 3D box alignment, closely matching the geometry implied by the scene.

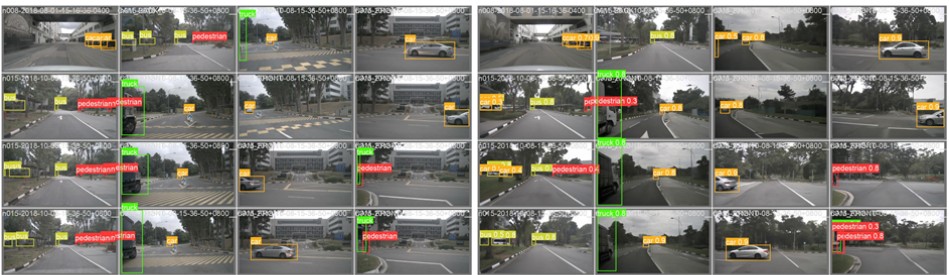

Real Video        Synthetic Video from DiVE

Figure 4: **Left:** Detection results on a real nuScenes image. **Right:** Detection results on a corresponding synthetic image generated by the DiVE baseline. Bounding boxes indicate detected objects (primarily vehicles).

### 4.3 Main Results: Improving Geometric Fidelity and Downstream Tasks

Table 2 shows RLGF's impact on the nuScenes validation set. RLGF consistently enhances geometric integrity (GeoScores) across baselines while maintaining or improving visual quality (FVD). For instance, DiVE + RLGF significantly improves all GeoScore components (VP error: 0.086 → 0.068; Lane F1: 0.792 →0.879; Depth RMSE: 1.822 → 0.772).

Crucially, this geometric enhancement translates to substantial 3DOD gains: DiVE + RLGF boosts mAP from 25.75 to 31.42 and NDS from 33.61 to 36.07, markedly closing the gap to real data performance. Similar improvements are seen for MagicDrive-v2 + RLGF (mAP: 18.95 → 23.21). These results underscore RLGF's effectiveness as a plug-and-play module for improving both geometric soundness and downstream utility of synthetic videos. We also present the qualitative result in fig. 3 about the detection results on our synthetic data and baseline [18] and the real data. (More qualitative visualization results and further analysis are included in the supplementary materials).

### 4.4 2D Object Detection Results

To illustrate that current diffusion models like DiVE can generate visually realistic data with minimal 2D domain gap for certain tasks, we present qualitative 2D object detection results. Figure 4 shows outputs from a YOLOv5 [20] detector applied to (a) real nuScenes data and (b) synthetic data generated by the DiVE baseline. The detector is pre-trained on a large-scale dataset (e.g., COCO) and then fine-tuned on real nuScenes training data.

As observed in Figure 4, the 2D detection performance on DiVE-generated synthetic data is qualitatively very similar to that on real data. Objects are generally detected with comparable confidence and bounding box accuracy. This visual consistency aligns with our quantitative findings (mAP: 43.8 on synthetic vs. 44.7 on real, as mentioned in the Introduction), suggesting that the semantic content and 2D appearance features necessary for 2D detection are well-preserved in the synthetic videos. This further reinforces our hypothesis that the primary limitation of such synthetic data lies in its 3D geometric fidelity, which is specifically addressed by our RLGF framework.

Table 3: **Performance of our Latent Occupancy Prediction Model ($\mathcal{P}_{occ}$) on the Occ3D-nuScenes.** $\mathcal{P}_{occ}$ predicts 3D occupancy grids from sequences of micro-decoded latent features. **Vehicle** and **Dri. Sur** indicate IoU of vehicles and driving surface.

| Method | Vehicle | Dri. Sur | mIoU ↑ |
|---|---|---|---|
| FlashOcc [70] | 43.2 | 72.2 | 32.08 |
| $\mathcal{P}_{occ}$ | 37.9 | 65.7 | 29.96 |

Table 4: **Ablation study of HGA reward components within RLGF on nuScenes.**

| ID | HGA Rewards | | | | | 3DOD | |
|---|---|---|---|---|---|---|---|
| | $r_{vp}$ | $r_{lane}$ | $r_{depth}$ | $r_{align}$ | $r_{iou}$ | mAP↑ | NDS↑ |
| DiVE [18] | | | | | | 25.75 | 33.61 |
| 1 | ✓ | | | | | 26.31 | 33.66 |
| 2 | ✓ | ✓ | | | | 26.93 | 33.98 |
| 3 | ✓ | ✓ | ✓ | | | 27.12 | 34.82 |
| 4 | | | | ✓ | ✓ | 28.06 | 35.11 |
| Full | ✓ | ✓ | ✓ | ✓ | ✓ | **31.42** | **36.07** |

Table 5: **Ablation studies on key RLGF hyperparameters.**

| Hyperparameter | Value | 3D Detection mAP |
|---|---|---|
| **Window Size (w)** | 3 | 30.89 |
| | **5 (Ours)** | **31.42** |
| | 8 | 31.25 |
| **Reward Weights ($\lambda$)** | Equal Weights (all 0.2) | 30.76 |
| | **Balanced (Ours)** | **31.42** |
| **Window Range Position ($t'$)** | Early (Random in [20, 30]) | 30.55 |
| | **Mid (Random in [8, 30]) (Ours)** | **31.42** |
| | Late (Random in [1, 15]) | 29.91 |

## 4.5 Ablation Study

### 4.5.1 Ablations about HGA Reward Components

We conduct ablation studies on nuScenes using DiVE as the baseline to understand the contributions of different HGA reward components. Results are shown in Table 4. Incrementally adding reward components ($r_{vp}$, $r_{lane}$, $r_{depth}$, and occupancy rewards ($r_{align} + r_{iou}$) progressively improves 3DOD performance (Table 4). For example, adding point-line-plane rewards (ID: 3) boosts DiVE's mAP to 27.12. Incorporating only occupancy rewards (ID: 4) yields 28.06 mAP. The full HGA system, combining all five components, achieves the highest mAP (31.42) and NDS (36.07), highlighting the synergistic benefits of our comprehensive multi-level feedback. (More qualitative visualization results and further analysis are included in the supplementary materials).

### 4.5.2 Ablations about Hyerparameters.

We provide the detailed ablation results in table 5 for clarity, including the RL window size, reward weights, and the starting step of the sliding window. All ablations are performed on the DiVE baseline and evaluated with BEVFusion.

## 5 Conclusion

In this work, we addressed the critical issue of geometric distortions in diffusion-based video generation for autonomous driving. We introduced GeoScores for quantitative evaluation and proposed Reinforcement Learning with Geometric Feedback (RLGF), a novel framework to enhance the geometric integrity of synthetic videos. RLGF, through its Hierarchical Geometric Alignment (HGA) module which incorporates multi-level geometric and scene occupancy feedback, effectively injects perception-driven constraints into the generation process by fine-tuning pre-trained diffusion models. Our experiments demonstrate that RLGF significantly improves geometric fidelity across multiple baselines and, crucially, boosts downstream 3D object detection performance by up to , substantially closing the gap with real-data performance. This work establishes a new direction for generating more reliable and task-aware synthetic data for autonomous systems.

## Acknowledgments

This work was supported by the Science and Technology Development Fund of Macau SAR (FDCT) under grants 0102/2023/RIA2 and 0154/2022/A3 and 001/2024/SKL, and the Jiangyin Hi-tech Industrial Development Zone under the Taihu Innovation Scheme (EF2025-00003-SKL-IOTSC).

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

# Appendix

This supplementary material provides additional details to support the findings presented in our main paper. We include: (1) comprehensive implementation specifics for our RLGF framework and the pre-trained perception models; further details on the GeoScores metric computation; (2) additional experimental results (3) a discussion on the limitations of our current work and potential future directions.

## A   Detailed Implementation Details

This section elaborates on the implementation details of our proposed RLGF framework, the pre-trained perception models ($\mathcal{P}_{geo}$ and $\mathcal{P}_{occ}$) , and the experimental setup.

### A.1   Dataset Preparation

All experiments, including the pre-training of our perception models ($\mathcal{P}_{geo}$ and $\mathcal{P}_{occ}$) and the fine-tuning of diffusion models with RLGF, are conducted using the nuScenes dataset [5]. We primarily utilize the official training and validation splits. While nuScenes provides rich annotations like 3D bounding boxes and HD maps, it does not directly offer ground truth labels for vanishing points (VP), dense segmentation masks for all relevant classes (like fine-grained lanes beyond HD map polylines), or per-pixel depth maps required by our $\mathcal{P}_{geo}$. Therefore, we generate high-quality pseudo-labels for these tasks using strong, pre-existing perception models, as detailed below. These pseudo-labels serve as the training targets for our latent-space perception models.

**Depth Pseudo-Labels:** To obtain dense depth information for training the depth estimation component of $\mathcal{P}_{geo}$, we utilize Depth Anything V2 (vit-l version) [65]. This state-of-the-art monocular depth estimation model is applied to all images in the nuScenes training set to generate per-pixel depth maps. These output depth maps serve as the pseudo-ground truth for our latent depth estimation task.

**Semantic Segmentation Pseudo-Labels (Lanes, Road Surface, Vehicles):** For precise segmentation masks of various scene elements, we employ Grounded-SAM-2[43, 42]. For the lanes, the model is prompted to accurately segment visible lane markings. The resulting binary segmentation masks are used as pseudo-ground truth for training the lane parsing head of $\mathcal{P}_{geo}$. For the road surface and vehicle masks, SAM-2 is also utilized to generate segmentation masks for road surfaces and vehicles.

**Vanishing Point Pseudo-Labels from Lane Masks, following [13]:** With accurate lane segmentation masks obtained via SAM-2 (as described above), we derive vanishing point pseudo-labels through a geometric procedure. For each detected lane marking in a frame: The center point of the lane marking is calculated from its left and right edges (derived from the SAM-2 segmentation mask) for every horizontal image line at 5-pixel intervals. These extracted center points are grouped to represent the centerline of each individual lane marking. Robust curve fitting (e.g., RANSAC with a line model) is applied to these centerlines. The intersection point of multiple fitted lane centerlines is then computed to determine the scene's vanishing point. This computed VP serves as the pseudo-ground truth for the VP detection task.

The use of these high-quality pseudo-labels enables us to train effective latent-space perception models tailored to the nuScenes domain, which subsequently provide the nuanced reward signals for our RLGF framework. The conditions $c$ for the main diffusion models (e.g., semantic 3D boxes for some baselines) are derived from the original nuScenes ground truth annotations.

### A.2   Perception Model Architectures and Pre-training

**Micro-Decode Module($\mathcal{F}_{micro}$):** The $\mathcal{F}_{micro}$ module is constructed using the first upper block of the official VAE decoder from the OpenSora [77] used by our baseline video diffusion models. The input of $\mathcal{F}_{micro}$ is the noisy latent feature $z_k^f$ for a frame $f$ with the timestep $k$. The same $\mathcal{F}_{micro}$ architecture is used when processing the reference real-video latent $z_v$ (with $k$ typically set to 0).

**Latent Geometry Perception Model($\mathcal{P}_{geo}$):** We use a pre-trained DINOv2-ViT-S/14 [36] as the backbone feature extractor and the pre-trained weight from DepthAnything-V2 [65]. $\mathcal{P}_{geo}$ is trained

for 50 epochs on the nuScenes training split using the AdamW optimizer with a learning rate of $5 \times 10^{-5}$ and a batchsize of 16. We use $8\times$ NVIDIA A100 GPUs to cover the experiment.

**Latent Occupancy Prediction Model($\mathcal{P}_{occ}$):** $\mathcal{P}_{occ}$ is trained on occ3D-nuscenes dataset for 24 epochs using AdamW optimizer following [70, 16].

### A.3   RLGF Fine-tuning Details

Baselines: We used publicly available checkpoints for MagicDrive-V2 [9] and DiVE [18].

LoRA Configuration: For LoRA, we applied it to the attention layers (Q, K, V projections) of the DiT backbone in the diffusion models. We used a rank $r = 16$ following [1].

Latent-Space Windowed Optimization: The window size $w$ is set to 5. The starting step $t'$ for the window was randomly sampled from the range $[8, 30]$, with $T = 30$ is the total number of diffusion steps.

Reward Weights: We set $\lambda_{vp} = 0.1, \lambda_{lane} = 0.1, \lambda_{depth} = 0.5$. These weights were determined empirically based on early experiments on a small validation subset, aiming to balance the scale of individual reward components and their perceived impact on generation quality.

We use AdamW with a learning rate of $1 \times 10^{-4}$ and a batchsize of 1 with 8 frames per video clip.

### A.4   GeoScores Metric Details

This section provides further clarification on the computation of our GeoScores components. For all GeoScores, the "reference ground truth" is derived by applying the corresponding pre-trained perception model to the *real* video data, following appendix A.1. The score then measures the deviation of the synthetic video's perception output from this real-data-derived reference.

**Vanishing Point Error (VP↓):** Calculated as the L2 Normalized Distance (NormDist) between the calculated VP on a synthetic frame and the VP calculated on a real frame. **Lane Topology Score (Lane↑):** Calculated as the F1 score for semantic segmentation of lane markings. The predictions is from Grounded-SAM2 [43, 42] on the synthetic frame, and the target is applied to the real frame. **Depth Error (Depth)↓:** Calculated as the Root Mean Squared Error (RMSE) between the depth map predicted by Depth Anything V2 [65] for road surface regions on a synthetic frame and the depth map for the same regions on the real frame. Road surface masks are obtained from SAM-2 [42].

## B   Additional Experiment Results

### B.1   Extended 3D Object Detection Results on Multiple Detecters

To further demonstrate the generalizability of the improvements conferred by RLGF, we evaluated the generated synthetic data using an additional state-of-the-art 3D object detector, StreamPETR [52], alongside the BEVFusion results presented in the main paper. Table 6 presents the performance (mAP and NDS on nuScenes validation) for StreamPETR and the average performance across both BEVFusion and StreamPETR. Both detectors were trained from scratch solely on the respective synthetic data or real data.

### B.2   Robustness to Different Downstream Detectors:

To demonstrate generality, we evaluated our method on the stronger **StreamPETR** detector. RLGF achieved a substantial +4.75% mAP gain, proving our geometric improvements are robust and benefit diverse downstream architectures.

## C   Limitations and Future Work

This section discusses the current limitations of our RLGF framework and GeoScores metric, alongside potential avenues for future research.

**Dependence on Perception Models:** RLGF's performance is inherently tied to the accuracy and robustness of the pre-trained perception models ($\mathcal{P}_{geo}, \mathcal{P}_{occ}$). Biases or errors in these models could

Table 6: Detailed 3D Object Detection (3DOD) performance on nuScenes validation using Stream-PETR [52]. RLGF is applied to MagicDrive-v2 and DiVE.

| Methods | Quality | BevFusion | | StreamPETR | |
|---|---|---|---|---|---|
| | FVD | mAP | NDS | mAP | NDS |
| Real Data | - | 35.53 | 41.20 | 38.01 | 49.02 |
| Panacea [57] | 139.0 | 11.58 | 22.31 | - | - |
| Drive-WM [57] | 122.7 | 20.66 | - | - | - |
| MagicDrive-v2 [18] | 101.2 | 18.95 | 21.10 | 22.77 | 28.93 |
| DiVE [18] | 68.4 | 25.75 | 33.61 | 29.19 | 36.23 |
| MagicDrive-v2+Ours | 99.8 | 23.21 | 27.80 | 26.01 | 35.64 |
| DiVE+Ours | **67.6** | **31.42** | **36.07** | **33.94** | **39.68** |

| Method | mAP (BEVFusion) |
|---|---|
| DiVE [16] (Baseline) | 25.75 |
| + Detector Reward | 26.51 (+0.76) |
| **+ Ours (RLGF)** | **31.42 (+5.67)** |

Table 7: Comparison of reward signal effectiveness on the DiVE baseline. Our Hierarchical Geometric Reward (HGR) is significantly more effective than a high-level detector-based reward.

propagate into the reward signal and mislead the generation process. Future work could explore jointly training or adapting perception models during RLGF, or using ensembles.

**Computational Cost:** While Latent-Space Windowed Optimization significantly reduces costs compared to full rollouts, RL-based fine-tuning remains more computationally intensive than standard diffusion model training. Exploring more sample-efficient RL algorithms or distillation techniques could be beneficial.

**Reward Design and Balancing:** The current HGA reward combines five components with manually tuned weights. Optimizing these weights automatically or learning a more adaptive reward function is a promising direction. Furthermore, incorporating even more diverse geometric or physical constraints (e.g., collision avoidance, traffic rule adherence) could further enhance realism.

**Generalization:** While demonstrated on nuScenes, further investigation is needed to assess RLGF's generalization capabilities across diverse datasets, environmental conditions (e.g., adverse weather, night scenes not well-represented in training), and different diffusion model architectures.

**GeoScores Scope:** Current GeoScores focus on camera-based geometric aspects. Expanding them to include LiDAR consistency or multi-modal geometric agreement could provide a more holistic evaluation.

