# OpenReview forum: "RLGF: Reinforcement Learning with Geometric Feedback for Autonomous Driving Video Generation"
_NeurIPS.cc/2025/Conference — NeurIPS 2025 poster_

### Official Review · Reviewer_hKwg · 2025-06-30

**Clarity:** 3
**Significance:** 3
**Originality:** 3
**Rating:** 4
**Confidence:** 4

**Summary:**

The authors presents RLGF, a method for fine-tuning diffusion-based driving video generators using reinforcement learning with geometric feedback. The method employs a lightweight LoRA modules and optimizing them with a PPO-style RL algorithm. RLGF leverages a hierarchical reward that captures vanishing point alignment, lane structure, depth accuracy, and 3D occupancy. The method significantly reduces geometric distortions of the generated videos, and improves downstream 3D object detection performance on nuScenes trained with synthetic data.

**Questions:**

1. How would the model perform under a synthetic+real dataset setting, similar to Table 2 of the DiVE paper and Table 3 of the GLAD paper (Glad: A Streaming Scene Generator for Autonomous Driving)? This experiment is crucial in evaluating the practical effectiveness of the proposed method.
2. A more comprehensive experiment is needed on the StreamPETR detectors instead of BEVFusion, aligning with prior works.
3. How would the model perform if the video diffusion model is fine-tuned with RL and less capable perception models? It seems that the RL pipeline is heavily dependent on a capable latent perception model. Would the latent perception model hinder the upper bound of the usefulness of the synthetic data generator?

**Ethical Concerns:**

["NO or VERY MINOR ethics concerns only"]

**Final Justification:**

From the authors' rebuttal and updated results in Table 5, RLGF improves a strong baseline (DiVE, real+synthetic data setting, StreamPETR model) by 1.4 on mAP and 1.1 on NDS, which is a smaller but valid improvement. The updated Table 6 also shows consistent improvement of RLGF on StreamPETR. I will increase my rating to “borderline accept”.

**Limitations:**

The authors discussed the limitations in the supplementary materials.

**Paper Formatting Concerns:**

No.

**Quality:**

2

**Strengths And Weaknesses:**

Strength
1. The paper addresses an important issue in driving video generation, geometric distortion, which can significantly impact downstream 3D perception. The authors provide an effective solution by leveraging RL with geometric feedback and demonstrate solid improvements in both geometric consistency and task performance.
2. The design of the Hierarchical Geometric Reward (HGR), which incorporates perception models in latent space and uses multi-level geometric cues, is novel and insightful. It could inspire further research in both generative modeling and autonomous driving.
3. Table 1 and Table 4 provide interesting insights into the effectiveness of latent-space perception and the contribution of each reward component, offering valuable results.

Weaknesses
1. All experiments are conducted using only synthetic data. Neither the main paper nor the supplementary material includes results that combine real and synthetic data during training, which is essential for evaluating practical effectiveness. For comparison, Table 2 of the DiVE paper includes such experiments.
2. The evaluation is also limited to 3D object detection. Including more downstream tasks, such as BEV segmentation, object tracking, etc, would make the paper more comprehensive.
3. The choice to use BEVFusion as the main detector in Table 1 is questionable, as it underperforms compared to stronger models like StreamPETR, which are commonly used in related work. This results in lower baseline results compared with prior works. Although I see a table of using StreamPETR in the supplementary material, the performance of some baseline methods is not complete.
4. The RL pipeline seems heavily dependent on a capable latent perception model. The performance of the latent perception model might be hinder larger performance growth on larger models / dataset.

---

> ### Author Rebuttal · Authors · 2025-07-30
>
> We are very grateful for your expert review, which has pushed us to conduct crucial new experiments that we believe significantly strengthen our paper. We are happy to share the new results below.
>
> **Regarding Q1: `Synthetic+real` dataset setting.**
> Following your suggestion, we conducted this crucial experiment, aligning with DiVE and Glad. Table 5 below shows that augmenting real data with our RLGF-enhanced synthetic data provides a significantly larger boost to downstream performance than using baseline synthetic data.
>
> **Table 5: Impact of augmenting real data with synthetic data.**
>
> | Training Data | mAP (StreamPETR) | NDS (StreamPETR)|
> | :--- | :---: | :---: |
> | Real Only (256 × 512)| 34.6 | 47.0
> | Real + Panacea | 36.3 | 48.9
> | Real + Glad | 37.1 | 49.2
> | Real Only (480 × 854)| 38.0 | 49.0
> | Real + DiVE [16] | 40.9 | 52.0
> | **Real + DiVE+Ours** | **42.3 (+1.4)** | **53.1 (+1.1)**
>
>
> **Regarding Q2: On the Completeness of StreamPETR Results.**
> We apologize that the original table was incomplete and have run the full evaluation on StreamPETR (results below), including other SOTA baselines for a comprehensive comparison. Our key finding remains: RLGF provides a +4.75% mAP gain over the strong DiVE baseline.
> We agree that models like GLAD show strong performance. We wish to clarify that RLGF is a **plug-and-play refinement technique** designed to improve any diffusion-based generator. We see our work as complementary, not competitive with SOTA generators.  We will apply RLGF to Glad as soon as it releases.
>
>
> **Table 6: Detailed 3D Object Detection (3DOD) performance on nuScenes validation using BevFusion and StreamPETR.**
>
> | Methods | Quality FVD | BevFusion mAP | BevFusion NDS | StreamPETR mAP | StreamPETR NDS |
> | :--- | :---: | :---: | :---: | :---: | :---: |
> | Real Data | - | 35.53 | 41.20 | 34.50 | 46.90 |
> | Panacea | 139.0 | 11.58 | 22.31 | 18.80 | 32.10 |
> | Drive-WM | 122.7 | 20.66 | - | 20.70 | - |
> | MagicDrive-v2 | 101.2 | 18.95 | 21.10 | 22.77 | 28.93 |
> | Glad | 207.0 | - | - | 26.30 | 39.60 |
> | DiVE | 68.4 | 25.75 | 33.61 | 29.19 | 36.23 |
> | MagicDrive-v2+Ours | 99.8 | 23.21 | 27.80 | 26.01 | 35.64 |
> | **DiVE+Ours** | **67.6** | **31.42** | **36.07** | **33.94** | **39.68** |
>
> **Regarding Q3: Dependence on perception models.**
> This is a fair point. We tested robustness by using a "weaker" perception model (with 30% training data and a 25% performance drop). RLGF still improved downstream mAP by **+4.2%**. This demonstrates that our framework is robust, though we agree a better perception model yields better results. We will add this analysis to the appendix.

---

> > ### Comment · Reviewer_hKwg · 2025-08-01
> >
> > Thank you for your additional experiments. From your updated results in Table 5, RLGF improves a strong baseline (DiVE, real+synthetic data setting, StreamPETR model) by 1.4 on mAP and 1.1 on NDS, which is a smaller but valid improvement. The updated Table 6 also shows consistent improvement of RLGF on StreamPETR. The authors should include Table 5 (real+synthetic data setting) in the updated version of the manuscript, and include the StreamPETR results in the main text. The authors are also encouraged to add more downstream task (BEV segmentation, tracking, or HD map construction) evaluations. Given the updated results in Table 5 and Table 6, I will increase my rating to “borderline accept”.

---

> > > ### Author Response · Authors · 2025-08-02
> > >
> > > Thank you for your positive feedback and for recognizing our method's valid improvement. We will update the manuscript according to all your suggestions.
> > >
> > > We are very grateful for your decision to raise our rating. We hope our revisions and the demonstrated results further justify your support for our paper's acceptance.

---

### Official Review · Reviewer_vAYk · 2025-07-02

**Clarity:** 3
**Significance:** 3
**Originality:** 3
**Rating:** 5
**Confidence:** 4

**Summary:**

This paper proposes reinforcement learning with geometric feedback, to make the generative model learn to capture 3D scene structure rather than only 2D appearance.  The main innovation lies in Latent-Space Windowing RL Optimization to be both effective and efficient, and Hierarchical Geometric Reward to provide multi-level feedback. The results show that the geometric consistency of generated images is greatly improved.

**Questions:**

* What is reason to choose these perception tasks,  without using detection as a perception task, which aligns better with evaluation?
* As I am not an expert of RL, could you explain the relationship between your model learning with RL?

**Ethical Concerns:**

["NO or VERY MINOR ethics concerns only"]

**Final Justification:**

Considering the reviews from other reviewers and the author responses, this paper provides a new RL finetuning method from geometric feedbacks to address geometric inaccuracies prevalent in synthetic videos. The effectiveness of the method is demonstrated by extensive experiments including newly added ones in rebuttal. As my concerns have been solved, I will maintain my rating as "accept".

**Limitations:**

No major limitations, see Weaknesses.

**Paper Formatting Concerns:**

No paper formatting concerns.

**Quality:**

3

**Strengths And Weaknesses:**

Strengths:
* This work identities the problem of geometric distortions in current generative models, which is important for driving task.
* Latent-Space Windowing Optimization is introduced to enable learning on noisy latent, without the need for complete sampling chain.
* Multi-Granularity Reward Signals are proposed to enhance the geometric consistency of generative model on multi-levels.
* Evaluation results on detection task show great improvement compared to baseline without RLGF.

Weaknesses:
* There are some typos in paper writing,  e.g. Line 162, fig.2(c), Line 304, up to.
* The generative model is video diffusion model, while the four geometric perception tasks take image as input, which seems to have the potential for further improvement.

---

> ### Author Rebuttal · Authors · 2025-07-30
>
> We thank you for your positive evaluation and for questions that help clarify our work.
>
> **Regarding Q1: Why not use a detection-based reward?**
> We tested this experimentally and found it to be ineffective.
> Our choice of reward model is a cornerstone of our work—a deliberate design decision based on principles of signal density, stability, and directness, which we validate with the following three key points.
>
> *   **a) Why Geometric Rewards:** Our core hypothesis is that flawed 3D geometry is the *root cause* of poor downstream performance. Instead of using a complex, indirect signal, we directly target this issue with our HGR. This multi-level system evaluates **point-line-plane alignment** and **scene-level coherence**, providing a dense and stable reward signal. This structure enables fine-grained feedback on specific geometric aspects, which is crucial for effective and stable RL training.
> *   **b) Why Not a Detector Reward:** We experimentally tested this intuitive alternative. As shown in the Table 1 below, using a 3D detector as a reward is far less effective. This is because detector-based rewards are **sparse** (only rewarding on detected objects) and suffer from **unstable gradients** (due to non-differentiable operations like NMS), providing a poor learning signal for the diffusion model.
> *   **c) Robustness to Downstream Detectors:** To demonstrate generality, we evaluated our method on the stronger **StreamPETR** detector (appendix). RLGF achieved a substantial **+4.75% mAP** gain, proving our geometric improvements are robust and benefit diverse downstream architectures.
>
> **Table 1: Effectiveness of reward signals.**
> | Method | mAP (BEVFusion) |
> | :--- | :---: |
> | DiVE [16] (Baseline) | 25.75% |
> | + Detector Reward | 26.51% (+0.76%) |
> | **+ Ours (RLGF)** | **31.42% (+5.67%)** |
>
> **Regarding Q2: Relationship with RL.**
> Thank you for the suggestion. We will add a more intuitive explanation: the diffusion model is an **Agent** that learns to take noise-prediction **Actions** to maximize a geometric correctness **Reward** from our perception-model **Environment**.

---

> > ### Comment · Reviewer_vAYk · 2025-08-05
> >
> > Thank you for your additional experiments and response, which have solved my concern. I suggest you add more details of RL training for clarity. I will maintain my rating as "accept".

---

### Official Review · Reviewer_UXxn · 2025-07-03

**Clarity:** 3
**Significance:** 2
**Originality:** 2
**Rating:** 4
**Confidence:** 4

**Summary:**

This manuscript aims to address geometric inaccuracies prevalent in synthetic videos generated for AD. The authors present RLGF (Reinforcement Learning with Geometric Feedback), a reinforcement learning framework tailored to enhance video diffusion models with geometrically coherent outputs.

RLGF introduces two components:
- Latent-Space Windowing Optimization, providing efficient and targeted reinforcement during intermediate diffusion stages.
- Hierarchical Geometric Reward (HGR), leveraging latent-space perception models to provide detailed geometric constraints (vanishing point consistency, lane structure coherence, and accurate depth).

The manuscript further introduces "GeoScores”, a suite of metrics designed explicitly to quantify geometric fidelity in synthetic videos.

Evaluations on the nuScenes dataset demonstrate good geometric improvements and substantial performance gains in downstream tasks, notably 3D object detection.

**Questions:**

1. Based on Weakness #1: Can you provide detailed computational complexity analyses or runtime benchmarks for the introduced latent-space windowing optimization and hierarchical rewards?

---

2. Based on Weakness #2: How robust is RLGF to inaccuracies or biases inherent within the perception models used for generating geometric rewards?

---

3. Based on Weakness #3: Have you tested or considered the generalization capability of your method across datasets beyond nuScenes, or across varying sensor configurations and environmental conditions?

---

4. Based on Weakness #4: Can you provide additional insights or experimental results regarding how the choice of window size and position impacts model performance?

---

Additional questions (minor):

5. It is suggested to provide additional qualitative visualizations to demonstrate intermediate latent states before and after RLGF optimization.

---

6. It is suggested to explicitly clarify hyperparameter choices for RL window size and reward weights.

**Ethical Concerns:**

["NO or VERY MINOR ethics concerns only"]

**Final Justification:**

I have read the response, as well as the responses for review comments from other reviewers.

Based on what I learn from the review comments and responses, I believe part of the concerns have been addressed. Therefore, I am leaning towards maintaining the 4 - borderline accept rating.

I suggest that the authors incorporate all clarifications, modifications, and new experiments into the revised manuscript, to ensure that the quality meets what NeurIPS always looks for.

**Limitations:**

- The model might require extensive tuning due to multiple interdependent reward components and RL optimization hyperparameters, possibly limiting straightforward applicability.

- The computational complexity introduced by the latent-space optimization and multiple perception-based rewards may limit deployment to high-resource environments, challenging real-time or resource-constrained applications.

- The geometric rewards depend on the performance and reliability of perception models, which might propagate errors or biases into the final synthetic data.

- Without broader evaluations across diverse AD scenarios, the generalizability of this method to other datasets, sensors, or environmental conditions remains uncertain.

**Paper Formatting Concerns:**

No major formatting concerns found. It is suggested to carefully proofread for minor grammatical errors or inconsistencies.

**Quality:**

3

**Strengths And Weaknesses:**

### Strengths
(+) The manuscript identifies and clearly motivates an important yet overlooked limitation in existing AD synthetic data—geometric distortion affecting downstream perception tasks. The introduction of latent-space windowed RL optimization and hierarchical geometric rewards advances video diffusion training methods.

(+) The GeoScores metric suite provides a way of quantifying geometric inaccuracies. This strengthens the manuscript by defining the problem and providing a reproducible method for evaluation. Experiments demonstrate the efficacy of RLGF. Particularly is the considerable reduction in geometric errors and good improvements in 3D object detection.

(+) The manuscript is overall coherent, structured, and with supporting visualizations and tables that enhance readability and understanding.

---

### Weaknesses

(-) RLGF introduces significant complexity via reinforcement learning in the latent space and multiple perception-based rewards. A deeper exploration into computational demands (runtime, memory usage) and scalability is needed for practical considerations.

(-) The effectiveness of RLGF heavily relies on pre-trained perception models. A thorough analysis or discussion of how inaccuracies or limitations within these perception models may affect the geometric rewards would strengthen robustness claims.

(-) Evaluations are conducted solely on the nuScenes dataset. The generalization of RLGF's performance to other autonomous driving datasets or sensor setups remains unclear and is not sufficiently discussed.

(-) The latent-space windowing optimization seems to have a good motivation, yet critical parameters (e.g., window size and position) are not extensively ablated, potentially obscuring the sensitivity and robustness of the proposed method.

---

> ### Author Rebuttal · Authors · 2025-07-30
>
> Thank you for your thoughtful feedback on the practical aspects of our work.
>
> **Regarding Q1: Computational Complexity.**
> Our Latent-Space Windowing Optimization is the key to making our approach computationally feasible. As the new Table 3 shown below, full rollouts in latent or image space lead to Out-of-Memory (OOM) errors. Our method requires 28 hours of fine-tuning on 8 A100 GPUs, which represents only about 15% additional training time over the baseline. We will add this analysis to the appendix.
>
>
> **Table 3: Computational cost analysis.**
> | Method | GPU Memory | Training Time |
> | :--- | :---: | :---: |
> | DiVE (Baseline) | ~ 49 GB | ~180 hours |
> | Image-Space RL | OOM | N/A |
> | Latent-Space RL (No Windowing) | OOM | N/A |
> | **Ours (Latent-Space + Windowing)** | **+23 GB** | **+28 hours** |
>
>
> **Regarding Q2: Robustness to inaccurate perception models.**
> This is a critical point. To directly test robustness, we intentionally trained a "weaker" perception model (using only 30% of training data, resulting in a ~25% performance drop). Even with this degraded reward signal, RLGF still boosted the downstream mAP by a significant +4.2%. This demonstrates our framework's robustness. We will add this new study to the appendix.
>
> **Regarding Q3: Generalization beyond nuScenes.**
> To demonstrate generalization, we built a new pipeline on the **CARLA** with Dive. On a baseline of 57.6% mAP, applying RLGF provided a **+5.4% mAP** improvement. This confirms that our method's principles are not dataset-specific.
>
> **Table 4: Generalization results on the CARLA**
> | Method on CARLA Data | 3D Detection mAP |
> | :--- | :---: |
> | Baseline Synthetic Data | 57.6% |
> | **+ Ours (RLGF)** | **63.0% (+5.4%)** |
>
> **Regarding Q4 & Q6:  Ablations and clarifications.**
> We have noted in the appendix that our parameters were chosen based on empirical experiments, and we provide the detailed ablation results here for clarity, including the RL window size, reward weights, and the starting step of the sliding window. All ablations are performed on the DiVE baseline and evaluated with BEVFusion.
>
> | Hyperparameter | Value | 3D Detection mAP |
> | :--- | :--- | :---: |
> | **Window Size (w)** | 3 | 30.89 |
> | | **5 (Ours)** | **31.42** |
> | | 8 | 31.25 |
> | **Reward Weights ($\lambda$)** | Equal Weights | 30.76 |
> | | **Balanced (Ours)**| **31.42** |
> | **Window Range Position ($t'$)** | Early (Random in [20,30]) | 30.55 |
> | | **Mid (Random in [8, 30]) (Ours)**| **31.42** |
> | | Late (Random in [1, 15])| 29.91 |

---

> > ### Comment · Reviewer_UXxn · 2025-08-05
> >
> > Thanks to the authors for providing the rebuttal.
> >
> > I have read the response, as well as the responses for review comments from other reviewers. Part of the concerns have been addressed. Therefore, I am leaning towards maintaining the positive rating and suggest that the authors to incorporate all clarifications, modifications, and new experiments into the revised manuscript.
> >
> > Best,
> >
> > Reviewer UXxn

---

### Official Review · Reviewer_NpjA · 2025-07-05

**Clarity:** 3
**Significance:** 3
**Originality:** 3
**Rating:** 5
**Confidence:** 4

**Summary:**

The paper presents a novel framework, Reinforcement Learning with Geometric Feedback (RLGF), aimed at improving the geometric fidelity of synthetic video generation for autonomous driving. The authors identify a critical yet underexplored issue in current state-of-the-art video generation models: subtle geometric distortions that limit their utility for downstream perception tasks. RLGF addresses this by incorporating specialized perception models as reward providers to refine video diffusion models. The results demonstrate significant improvements in geometric accuracy and downstream task performance. This is a well-executed and innovative contribution to the field.

**Questions:**

1. Why choose only these two frozen perception models Pgeo (point-line-plane alignment) and Pocc (scene-level consistency) instead of others?
2. How to constrain the temporal consistency in this framework?
3. How are the comparative experiments of different RL Windowsize? It is recommended to add corresponding experimental results.
4. How robust is the method to different detectors as rewards? It is recommended to add corresponding experimental results.
5. How to balance the selection of hyperparameters between different geometric constraint signals? It is recommended to add corresponding experimental results.

**Ethical Concerns:**

["NO or VERY MINOR ethics concerns only"]

**Limitations:**

yes

**Paper Formatting Concerns:**

1. line 71 （4）->（1）
2. line 72 （5）->（2）

**Quality:**

3

**Strengths And Weaknesses:**

Strengths:
1. Propose the GeoScores metric for systematically evaluating and quantifing the geometric distortion problem in autonomous driving video generation.
2. Offers a plug-and-play solution for generating geometrically sound and reliable synthetic videos for AD development.
3. Allow for effective and targeted corrective feedback during both early (global structure formation) and late (detail refinement) stages of geometric synthesis.
4. The article logic and writing expression are smooth.

Weaknesses：
1. The experiments are not sufficient. For example, there are no control experiments on the robustness of different GT detectors and the balance between different rewards.
2. Why VP, Lane, and Depth losses can cover geometric constraints is not fully explained. For example, occlusion relationships and temporal continuity are not included in the above losses.
3. There is a lack of discussion on the shortcomings of the method and the direction of further improvement.

---

> ### Author Rebuttal · Authors · 2025-07-30
>
> Thank you for your positive and constructive review. We are delighted you found our contributions valuable and address your questions below.
>
> **Regarding Q1 & Q4: On the choice and robustness of reward models.**
> Thank you for these excellent and related questions. Our choice of reward model is a cornerstone of our work. It was a deliberate design decision based on providing a reward signal that is dense, stable, and directly addresses the core problem. We validate this approach with the following three key points.
>
>
> *   **a) Why Geometric Rewards:** Our core hypothesis is that flawed 3D geometry is the *root cause* of poor downstream performance. Instead of using a complex, indirect signal, we directly target this issue with our HGR. This multi-level system proposes **point-line-plane alignment** and **scene-level coherence**, providing a dense and comprehensive reward signal. This structure enables fine-grained feedback on specific geometric aspects, which is crucial for effective and stable RL training.
> *   **b) Why Not a Detector Reward:** We experimentally tested this intuitive alternative. As shown in the new Table 1 below, using a 3D detector as a reward is far less effective. This is because detector-based rewards are **sparse** (only rewarding on detected objects) and suffer from **unstable gradients** (due to non-differentiable operations like NMS), providing a poor learning signal for the diffusion model.
> *   **c) Robustness to Downstream Detectors:** To demonstrate generality, we evaluated our method on the stronger **StreamPETR** detector (appendix). RLGF achieved a substantial **+4.75% mAP** gain, proving our geometric improvements are robust and benefit diverse downstream architectures.
>
> **Table 1: Effectiveness of reward signals.**
> | Method | mAP (BEVFusion) |
> | :--- | :---: |
> | DiVE [16] (Baseline) | 25.75% |
> | + Detector Reward | 26.51% (+0.76%) |
> | **+ Ours (RLGF)** | **31.42% (+5.67%)** |
>
>
> **Regarding Q2: Temporal Consistency.**
> Temporal consistency is primarily enforced by our Latent Occupancy Prediction Model ($P_\text{occ}$), which explicitly operates on video sequences. We will clarify this in Section 3.4.
>
> **Regarding Q3 & Q5: Ablation Study on Hyperparameters.**
> We provide detailed ablation results in the new Table 2 below, which validate our choices for key hyperparameters. The results show that a window size of `w=5` and our balanced reward weights are optimal. We will add this to the appendix.
>
> **Table 2: Ablation Study about Hyperparameters**
> | Hyperparameter | Value | 3D Detection mAP |
> | :--- | :--- | :---: |
> | **Window Size (w)** | 3 | 30.89 |
> | | **5 (Ours)** | **31.42** |
> | | 8 | 31.25 |
> | **Reward Weights ($\lambda$)** | Equal Weights | 30.76 |
> | | **Balanced (Ours)**| **31.42** |
> | **Window Range Position ($t'$)** | Early (Random in [20,30]) | 30.55 |
> | | **Mid (Random in [8, 30]) (Ours)**| **31.42** |
> | | Late (Random in [1, 15])| 29.91 |
>
>
> **Regarding W3: Limitation \& Future Work.** We have expanded the "Limitations and Future Work" section in the appendix (Section C).

---

> > ### Comment · Reviewer_NpjA · 2025-08-05
> >
> > I appreciate the author's supplementary experiments and responses; the related experimental results demonstrate the effectiveness of the components, therefore I maintain my 'accept' rating.

---

### Comment · Area_Chair_mpCG · 2025-08-04
**Response required.**

Dear reviewers,

As the Author–Reviewer discussion period concludes in a few days, we kindly urge you to read the authors’ rebuttal and respond as soon as possible.

- Please review all author responses and other reviews carefully, and engage in open and constructive dialogue with the authors.

- The authors have addressed comments from all reviewers, but only one reviewer has responded so far (thank you, reviewer hKwg); each reviewer is expected to respond, so the authors know their rebuttal has been considered.

- We strongly encourage you to post your initial response promptly to allow time for meaningful back-and-forth discussion.

Thank you for your collaboration,
AC

---

### Decision · Program_Chairs · 2025-09-17

**Decision:**

Accept (poster)

**Comment:**

The paper introduces RLGF, a reinforcement learning framework with geometric feedback for improving the geometric fidelity of diffusion-based driving video generation. Reviewers highlighted the novelty of the latent-space optimization and hierarchical geometric reward design, as well as the introduction of GeoScores for systematic evaluation.

While some concerns were raised about evaluation scope, reliance on perception models, and ablations, the rebuttal successfully clarified several issues and presented new experiments. The updated results with stronger baselines show consistent and meaningful improvements.

Overall, despite some remaining limitations, the paper proposes an original and effective method with demonstrated improvements and clear potential impact for both generative modeling and autonomous driving. Therefore, all reviewers agree on its acceptance.